# OpenReview forum: "Abstract Counterfactuals for Language Model Agents"
_NeurIPS.cc/2025/Conference — NeurIPS 2025 poster_

### Official Review · Reviewer_X1N5 · 2025-06-30

**Clarity:** 3
**Significance:** 4
**Originality:** 2
**Rating:** 4
**Confidence:** 2

**Summary:**

The paper works on the issue of counterfactual generation of language models at the token level. The issues arise because token-level actions might not be able to capture semantic meaning at a high-level. This might worsen in different environment. The paper works on this issue by proposing the idea of abstraction for an action from language models. The proposed method is model-agnostic, treating LM as a black-box model.

At high level, the propsoed method learns the abstraction concepts from the factual actions from the LM. Through the abstraction, the fraemwork generates the counterfactual profile in the abstraction space. Then it is mapped back the action space as the counterfactual action.

The proposed method is done by introducing another SCM that depends on observable, unobservable variables and actions. The support space of abstaction can be learned through either supervised or unsupervised setting.

The experimental section demonstrates the effectiveness of the proposed method on three benchmark datasets. The blackbox models are GPT2-XL and LLama3.2-1B. The author cliam that the token-level prediction tends to predict the observed token, while the abstaraction approach tends to predict the actual counterfactual action at the high level.

**Questions:**

1. How difficult is it to learn the abstraction space? In the paper, the abstraction can be learned in supervised or unsupervised setting. How is the support space defined in each case?
2. Is $f_y$ shared for every single $t$? If so, how do you make sure that the support of Y is consistent across different $t$? If not, does that mean we need to learn each independent SCM for every $t$?

**Ethical Concerns:**

["NO or VERY MINOR ethics concerns only"]

**Final Justification:**

I have read the comments from the authors and updated my score accordingly.

**Limitations:**

The only limitation that I can see is that only two LLM models are compared, which lack of convincing evidence for the claim made in the paper.

**Quality:**

3

**Strengths And Weaknesses:**

Strengths
- The issue that the paper is working on is significant, allowing the LM to be more explainable and transparent.
- The proposed method seems effective from the experimental results.
- The proposed method is fairly intuitive and easy to follow.

Weaknesses
- The main concern that I have is the novelty. It seems that the key contribution is the additional SCM that models the abstraction variable. - The intuition of why the abstraction variable works is not fully explained.
- Only two models are compared. Adding some of the recent models might be more convincing.

---

> ### Author Rebuttal · Authors · 2025-07-30
>
> >The main concern that I have is the novelty. It seems that the key contribution is the additional SCM that models the abstraction variable. - The intuition of why the abstraction variable works is not fully explained.
>
> Our main contribution is the introduction of an SCM which models the abstraction explicitly, and a method for generating counterfactual actions while performing counterfactual inference at the abstraction level. This allows us to perform counterfactual inference in a more meaningful manner where actions are identified with sequences of tokens, and whose meaning is context-dependent and may differ between factual and counterfactual settings.
>
> For example, suppose the model sees the prompt “He studied hard to become a” and generates the completion “journalist”. If we now intervene to change the prompt to “She is a”, the token-level exogenous posterior associated with the next token would be that of “to” (the fourth token in both sentences, which will have no support in the counterfactual context), rather than the profession. We want to define the exogenous factor over the high-level meaning of the sentence (e.g., profession), not the dynamics of the individual tokens.
>
> Our approach addresses this by abstracting the state-action pair into a higher-level variable $Y$, which captures semantic meaning independent of specific surface forms. We can then derive a posterior over the exogenous noise at the abstract level, allowing for meaningful counterfactual reasoning even when the context changes. This abstraction preserves interpretability and semantic consistency across factual and counterfactual scenarios.
>
> >How difficult is it to learn the abstraction space? In the paper, the abstraction can be learned in supervised or unsupervised setting. How is the support space defined in each case?
>
> In the supervised case, the support of the abstraction variable is defined by the user (explicitly, or based on available labelled datasets). Predicting the abstraction then becomes a traditional classification task, whose performance will depend on factors like data availability and quality.
>
> In the unsupervised case, the procedure is broken down in two steps, both consisting of auxiliary language model calls. Firstly, we define the support of the random variable Y (using first of the prompts in appendix F). Then, we can use the result of this step to classify arbitrary strings over this support, using another language model call (second prompt in appendix F).
>
> >Is $f_y$ shared for every single $t$? If so, how do you make sure that the support of $Y$ is consistent across different $t$? If not, does that mean we need to learn each independent SCM for every $t$?
>
> The mechanism $f_y$ remains the same across timesteps. In the supervised setting it is the modeller’s responsibility to ensure $Y$’s support is reasonable and spans the full range of relevant concepts. In practice, this set of concepts is usually easy to determine (e.g. professions for biography generation tasks). In the unsupervised setting, an auxiliary LLM call is used to discover the relevant concepts. For this, it is important to show this LLM a sufficient number of representative samples.
>
> >The only limitation that I can see is that only two LLM models are compared, which lack of convincing evidence for the claim made in the paper.
>
> Thank you for your comment. For easier comparison with previous work, in the text generation setting, we used the same models (and intervention checkpoints) as Ravfogel et. al., the token-level baseline. Furthermore, in the text-based games setting (MACHIAVELLI), we implement the policy using the OLMo-1B model.
>
> Our method is not tied to any specific implementation detail of any of the language models, and, by virtue of its black-box nature, is designed to work with any policy which can be easily sampled, combined with an abstraction.

---

> ### Author Response · Authors · 2025-08-05
>
> We thank the reviewer for their insightful comments. In our response, we have addressed the concerns and questions raised. Are there any other aspects you would like us to clarify?

---

### Official Review · Reviewer_NR3E · 2025-07-02

**Clarity:** 3
**Significance:** 3
**Originality:** 3
**Rating:** 5
**Confidence:** 4

**Summary:**

The paper proposes modeling abstract counterfactuals building on the use of gumbel counterfactuals for autoregressive language models. Compared to the prior method the abstract counterfactuals are categorically valued and dependent on sampled sequences, rather than counterfactual at the step wise token level. A structural causal model is given, an inference procedure introduced and it shown to be consistent. Empirical evaluations are done on three datasets using three suggested measures. The results show that the abstrat counterfactuals change the original distribution better than only relying on token level counterfactuals.

**Questions:**

* Do you feel the framing around language model agents is necessary?
* Do you see a way to sample from the models constrained on the abstract counterfactual?

**Ethical Concerns:**

["NO or VERY MINOR ethics concerns only"]

**Final Justification:**

I stand by my original review and am pleased with the author's response.

**Limitations:**

Yes

**Quality:**

4

**Strengths And Weaknesses:**

Strengths
* The point of needing higher level counterfactuals than the token based autoregressive ones is intuitive
* The paper is well written and clear
* The three step inference approach is clear
* Nice evaluation metrics

Weaknesses
* The preliminaries might need a slight bit of tightening, see smaller notes below

Notes
* The text in figure 1 is hard to read when printed
* Colors in figures were flat gray when printed
* I am not sure the use of numbers to indicate the actions in figure 1 and the introduction is better than typing out the action names in some typeface.
* Some might want to include the endogenous variables in the SCM tuple, and mention the exogenous ones are independent
* You might want to mention why the equation in line 100 is always well defined
* In line 105 you say LMs are always autoregressive, this is not correct.
* In line 109, K is not defined until line 117.
* The type of the functions in equation 1 and 2 does not strictly match the preliminaries, should the parents be clarified?

---

> ### Author Rebuttal · Authors · 2025-07-30
>
> >The text in figure 1 is hard to read when printed. Colors in figures were flat gray when printed
>
> Thanks for pointing this out. we will fix it in the camera-ready version.
>
> >I am not sure the use of numbers to indicate the actions in figure 1 and the introduction is better than typing out the action names in some typeface.
>
> We used numbers to indicate actions rather than typing them out to show them from the same perspective of the models, highlighting the ambiguity that results from the same symbols (numbers) being associated with different actions across the factual and interventional settings.
>
> One of the weaknesses of the token-level approach is precisely that the conditioning is done over the tokens output by the policy (the number, or action id) rather than the action’s meaning or full representation.
>
>
> >Some might want to include the endogenous variables in the SCM tuple, and mention the exogenous ones are independent
>
> Thanks for the suggestion, we will include the variables in the definition of the SCM for the camera ready version of the paper.
>
> >You might want to mention why the equation in line 100 is always well defined
>
> Thank you for the suggestion, we will explain that we assume that all $P(\mathbf{X}=\mathbf{x}) > 0$ for all observations.
>
>
> >In line 105 you say LMs are always autoregressive, this is not correct.
>
> Thank you for pointing this out. We will correct it in the camera-ready version of the paper.
>
> >In line 109, K is not defined until line 117.
>
> Thanks for noticing this. We will define the notation in line 108.
>
> >The type of the functions in equation 1 and 2 does not strictly match the preliminaries, should the parents be clarified?
>
> Thank you for your comment. We will re-order the arguments in the structural assignments displayed in these equations to match more closely the notation outlined earlier in the paper.
>
> >Do you feel the framing around language model agents is necessary?
>
> We find that the LM agent formalisation is more flexible, and encompasses the text-generation setting by capturing multi-token generation steps as a single action. We explain this connection in section 4.2. While it is true that most of the case studies we consider are concerned with the special case of text-generation, our method is more general and can be applied beyond these settings, hence our choice to use the more general LM agent framework.
>
> >Do you see a way to sample from the models constrained on the abstract counterfactual?
>
> This is an interesting point.
>
> Our approach corresponds to a form of rejection sampling based on the samples’ consistency with the counterfactual abstraction distribution. We constrain samples on the abstract counterfactual ‘a posteriori’.
>
> However, we think that constraining the language generation process in a direct / a-priori manner may not be feasible. One possible approach to this would require some sort of ‘model inversion’, and a probabilistic latent space. The abstraction could be defined as the embedding vector of the observation itself, with associated distribution in the associated probabilistic embedding space. Then, a counterfactual sample would be taken in this embedding space, which would be inverted back into text representation.
>
> There would be several difficulties with this approach, however. Firstly, we are not aware of any (large) language models that operate on a probabilistic embedding space, especially when considering representations of whole sentences. This would also limit our method to white-box models or models for which the providers allow access to their embeddings, which isn’t very common.
>
> Secondly, mapping backwards from an embedding to the corresponding generating text generally requires access to several supervised pairs of embeddings and prompts, and usually involves training a soft-token prefix for a frozen white-box model (e.g. GPT2) such that the completion matches the supervised un-embedded text. Even ignoring the fully deterministic nature of the embeddings used in these methods, the overall approach seems unlikely to yield text of the same quality as that produced by an arbitrary LLM.

---

> > ### Comment · Reviewer_NR3E · 2025-08-01
> >
> > Thanks for the responses, looking forward to seeing the final paper.

---

### Official Review · Reviewer_SCN4 · 2025-07-02

**Clarity:** 3
**Significance:** 3
**Originality:** 3
**Rating:** 4
**Confidence:** 3

**Summary:**

The paper introduces Abstract Counterfactuals (ACF), which is a framework for generating counterfactuals from LLM agents.
Previous approaches used token level counterfactuals, which didn't capture the semantic aspects of the LLM agent's actions.

The main contribution is the introduction of a semantic-level abstraction variable, Y, that captures the context and meaning, e.g., intent, profession, or emotion. As Y depends on the (factual) state and action, it is used for the counterfactual reasoning to obtain a counterfactual action that is semantically consistent with the factual action. The paper offers supervised and unsupervised methods to obtain Y.

The paper does experiments on ACF: a text-based game (MACHIAVELLI), a biography generation task, and emotion tracking in Reddit comments. Results show that ACF creates more semantically consistent counterfactuals than previous token-level approaches.

**Questions:**

In Appendix A, Proof of Proposition 1, the paper states: "We want to show that the expectation of our counterfactual distribution with respect to the exogenous values distribution is equal to the interventional distribution."
- How does the proof differs from the known result in causal inference that the expectation over the counterfactuals (individual level)  is the interventional probability.

-  What is an estimate for the time complexity of the sampling method?

- Are there formal approaches to know how coarse or fine-grained the abstractions are?

**Ethical Concerns:**

["NO or VERY MINOR ethics concerns only"]

**Final Justification:**

I'm satisfied with the authors' answers. No change of score from my side.

**Limitations:**

Nothing to add to the limitations the paper mentions

**Quality:**

3

**Strengths And Weaknesses:**

Quality
Strengths:
- The paper provides a well-defined approach based on structural causal modeling and counterfactual reasoning
- The paper validates the theory by doing experiments on different benchmarks
- The paper sets up evaluation metrics suitable to the counterfactual semantic consistency: Abstraction Change Rate, Counterfactual Probability Increase Rate, and Semantic Tightness.

Weaknesses:
- The creation of the abstraction Y depends on dataset, classifiers used, or results from unsupervised methods.
- The computational costs (e.g., sampling) could be challenging for practical real-world scenarios

Clarity
Strengths:
- The paper is clearly written and well-structured. The reader can easily follow the exposition for the SCM and the LLM Agent operating in a sequential decision-making environment.
- The examples are intuitive, e.g., the bear and lion scenario
- The details in the appendix are helpful

Weaknesses:
- Appendix B Metrics could benefit from more details

Significance
Strengths:
- The paper addresses issues in token-level counterfactual reasoning of LLM agents
- The ACF approach can be impactful for LLM agent safety and fairness

Weaknesses:
- The quality of the abstraction Y may vary in real-world scenarios as it depends on many factors

Originality
Strengths:
- The paper presents a new method, ACF, that is different from past approaches, e.g., token-level counterfactual reasoning
- The use of structural causal modeling with semantic abstraction Y for counterfactual reasoning is original.

Weaknesses:
- The creation of the abstraction

---

> ### Author Rebuttal · Authors · 2025-07-30
>
> >How does the proof differs from the known result in causal inference that the expectation over the counterfactuals (individual level) is the interventional probability.
>
> The proof mirrors the standard result in causal inference, which states that the expectation over individual-level counterfactuals recovers the interventional distribution. What we're doing here is adapting that proof to our specific method of computing counterfactuals. In essence, we're verifying that our approach preserves those results.
>
> >What is an estimate for the time complexity of the sampling method?
>
> Thank you for the question. The overall time complexity of our method is linear in the number of sampled realizations $N$. Each realization involves one forward pass through a language model, which has complexity $O(L \cdot d \cdot n^2)$, where $L$ is the number of layers, $d$ the hidden size, and $n$ the sequence length.
>
> Sampling from the latent space is efficient and negligible in comparison. We can use standard concentration bounds in principle to determine the number of realizations $N$ that achieves the desired statistical guarantees.
>
> >Are there formal approaches to know how coarse or fine-grained the abstractions are?
>
>
> Thank you for the question. One obvious metric for abstraction granularity is the number of abstraction classes; more classes typically indicate a finer-grained abstraction. However, this alone may not be sufficient, as many classes might be rarely or never used.
>
> A more informative measure is the mutual information between $Y \mid S$ and $A \mid S$. High mutual information suggests that the abstraction $Y$ retains a strong statistical relationship with the original action $A$, meaning $A$ can be nearly recovered from $Y$. In contrast, a coarser abstraction will lead to lower mutual information, as multiple actions will map to the same abstract representation.
>
> We mention this in the paper in section 3.2, line 232, when referring to the need for “statistical dependency” between the two.

---

> > ### Comment · Reviewer_SCN4 · 2025-08-05
> >
> > I thank the authors for clarifying my doubts. No further questions from my side. I keep my score.

---

### Official Review · Reviewer_5RqV · 2025-07-03

**Clarity:** 4
**Significance:** 3
**Originality:** 3
**Rating:** 5
**Confidence:** 4

**Summary:**

This paper proposes a framework for counterfactual text generation in language models. The main idea is that instead of performing counterfactual inference at the token or syntactic level, the abduction step in Judea Pearl's three-step counterfactual inference is carried out at a more abstract, semantical level. Experimental results show that on several benchmarks, the proposed method improves semantical consistency across factual and counterfactual contexts, over the token-level counterfactual generation methods.

**Questions:**

In addition to the questions raised earlier in connection with "weaknesses", another puzzling point is the claim that the proposed approach can treat a LLM as a black box. It makes sense for interventions that only manipulate prompts. But why is this the case when the interventions also manipulate the decoder \lambda?

**Ethical Concerns:**

["NO or VERY MINOR ethics concerns only"]

**Final Justification:**

The authors' responses to my concerns are reasonable. My evaluation remains positive after taking into account the authors' feedback and other reviews.

**Limitations:**

Depending on the authors' responses to the questions raised earlier, the section on limitations may need to be expanded or adjusted.

**Paper Formatting Concerns:**

None.

**Quality:**

3

**Strengths And Weaknesses:**

Strengths:

1. The motivations of the approach are sensible and interesting, and the proposal is quite novel

2. The technical presentation is largely clear and cogent.

3. The experimental results are interesting and fairly convincing.

Weaknesses:

1. It is not entirely clear if the alleged flaws of the token-level counterfactual generation are due specifically to the Gumbel-Max setup or reflect general problems with token-level counterfactuals. For example, one problem highlighted by the authors is that "TLCFs always increase the counterfactual probability of the observed token x^k", but the argument for this claim is specific to Gumbel-Max SCMs, and for general SCMs, it is not true that observing a certain value will inevitably increase this value's counterfactual probability. This seems to indicate that the problem is confined to Gumbel-Max models. On the other hand, however, the authors also tried to motivate their work by noting the general phenomenon of context-dependence of meaning, which suggests that the work is not just intended to improve counterfactual generation with Gumbel-Max models. A clarification of this issue would be appreciated.

2. Related to this ambiguity, and more importantly, if the claim is that there is a general problem with token-level counterfactuals going beyond the particular Gumbel-Max setup, the present paper does not yet provide a compelling demonstration of this general problem. In particular, it is unclear to me why f_A in general cannot be expected to properly represent the dependence of actions/generated texts on states/prompts that take into account the context-dependence of meaning. (Is the thought that we need an abstraction mechanism f_Y to tease out some context-invariant component of A?)

3. On p. 5, in step 2 "Counterfactual inference of Y'", it appears to require a joint probability law of U_A and U'_Y. It is unclear how the joint distribution is supposed to be determined? In particular, are they assumed to be independent, even though U'_Y is updated?

---

> ### Author Rebuttal · Authors · 2025-07-30
>
> >It is not entirely clear if the alleged flaws of the token-level counterfactual generation are due specifically to the Gumbel-Max setup or reflect general problems with token-level counterfactuals. For example, one problem highlighted by the authors is that "TLCFs always increase the counterfactual probability of the observed token x^k", but the argument for this claim is specific to Gumbel-Max SCMs, and for general SCMs, it is not true that observing a certain value will inevitably increase this value's counterfactual probability. This seems to indicate that the problem is confined to Gumbel-Max models. On the other hand, however, the authors also tried to motivate their work by noting the general phenomenon of context-dependence of meaning, which suggests that the work is not just intended to improve counterfactual generation with Gumbel-Max models. A clarification of this issue would be appreciated.
>
>
> Thank you for the thoughtful comment. You're right that the argument we presented regarding the increased counterfactual probability of the observed token under TLCFs is specific to Gumbel-Max Structural Causal Models (SCMs).
> However, our broader motivation extends beyond the Gumbel-Max framework. The main issue we aim to highlight is that token-level counterfactuals, regardless of the SCM instantiation, are tightly bound to the syntactic form of the model and often fail to reflect the underlying semantic dependencies in language.
>
> We will revise the manuscript to clarify this distinction between model-specific analysis and general motivation.
>
>
> >Related to this ambiguity, and more importantly, if the claim is that there is a general problem with token-level counterfactuals going beyond the particular Gumbel-Max setup, the present paper does not yet provide a compelling demonstration of this general problem. In particular, it is unclear to me why f_A in general cannot be expected to properly represent the dependence of actions/generated texts on states/prompts that take into account the context-dependence of meaning. (Is the thought that we need an abstraction mechanism f_Y to tease out some context-invariant component of A?)
>
>
> Thank you for the insightful comment. It is true that in general, $f_A$ does reflect context dependence, since it is implemented by a language model that sees the full prompt or preceding sequence. However, the issue arises in how counterfactual outcomes are computed. In standard token-level approaches, the 'abduction' step defines the exogenous factors over specific observed tokens. While these tokens make sense in the original context, their meaning can become ill-defined or unsupported in a different, counterfactual prompt.
>
> For example, suppose the model sees the prompt “He studied hard to become a” and generates the completion “journalist”. If we now intervene to change the prompt to “She is a”, the token-level exogenous posterior associated with the next token would be that of “to” (the fourth token in both sentences, which will have no support in the counterfactual context), rather than the profession. We want to define the exogenous factor over the high-level meaning of the sentence (e.g., profession), not the dynamics of the individual tokens.
>
> Our approach addresses this by abstracting the state-action pair into a higher-level variable $Y$, which captures semantic meaning independent of specific surface forms. We can then derive a posterior over the exogenous noise at the abstract level, allowing for meaningful counterfactual reasoning even when the context changes. This abstraction preserves interpretability and semantic consistency across factual and counterfactual scenarios.
>
> We will revise the paper to better articulate this limitation of token-level conditioning and to include an example like the one above for clarity.
>
> >On p. 5, in step 2 "Counterfactual inference of Y'", it appears to require a joint probability law of U_A and U'_Y. It is unclear how the joint distribution is supposed to be determined? In particular, are they assumed to be independent, even though U'_Y is updated?
>
> Thank you for raising this point.
>
> In brief, $U'_Y$ is determined in the abduction step by conditioning $U_Y$ on the observed state $S$ and abstraction $Y$. That is, we compute a posterior $p(U_Y \mid S, Y)$, and sample $U'_Y$ from this distribution. In Step 2, we then obtain the counterfactual abstraction $Y'$ by evaluating the combined mechanism $f_Y \circ f_A$ at the intervened state $S'$, using $U'_Y$ in place of $U_Y$. This allows us to propagate the context-dependent semantic content of the original observation into the counterfactual setting.
>
> Regarding the joint distribution: Following from the structure of our causal diagram, $U_A$ and $U_Y$ are independent a priori. Since only $U_Y$ is updated during abduction, we treat $U_A \sim p(U_A)$ as independent of $U'_Y \sim p(U_Y \mid S, Y)$. This results in an effective joint distribution $p(U_A) \cdot p(U'_Y \mid S, Y)$ used during the counterfactual rollout. We will revise the manuscript to clarify this assumption and make the mechanics of the abduction and inference steps more explicit.
>
> >In addition to the questions raised earlier in connection with "weaknesses", another puzzling point is the claim that the proposed approach can treat a LLM as a black box. It makes sense for interventions that only manipulate prompts. But why is this the case when the interventions also manipulate the decoder $\lambda$?
>
>
> Yes, when we say our method is black-box, we refer to the method of computing counterfactuals, given arbitrary interventions performed to the models (whether black-box or white-box). We study latent space interventions (gender steering) specifically, as they were interesting case studies presented in works we build upon.

---

> > ### Comment · Reviewer_5RqV · 2025-08-06
> > **Thanks**
> >
> > I appreciate the authors' responses to my comments. After reading all reviews and responses, my evaluation remains positive.

---

### Decision · Program_Chairs · 2025-09-17

**Decision:**

Accept (poster)

**Comment:**

This paper proposes Abstract Counterfactuals for language model agents, intervening at a higher action or interaction level rather than at the token level. The idea is well motivated, with clear potential to make counterfactual reasoning more semantically meaningful for agents. The writing is generally clear and the empirical results are encouraging across the reported tasks.

I believe the conceptual scope could be made clearer. The paper should more precisely define what counts as an abstract counterfactual and clearly separate its claims from token-level counterfactual methods, including a worked non Gumbel-Max example. It should also test the sensitivity of results to how the abstract variables are constructed and report failure cases.